# Impact of medication therapy management service on selected clinical and humanistic outcomes in the ambulatory diabetes patients of Tikur Anbessa Specialist Hospital, Addis Ababa, Ethiopia

**Zenebe Negash**[1]\*, **Alemseged Beyene Berha**[1], **Workineh Shibeshi**[1], **Abdurezak Ahmed**[2], **Minyahil Alebachew Woldu**[1], **Ephrem Engidawork**[1]

**1** Department of Pharmacology and Clinical Pharmacy, School of Pharmacy, College of Health Sciences, Addis Ababa University, Addis Ababa, Ethiopia, **2** Department of Internal Medicine, School of Medicine, College of Health, Sciences Addis Ababa University, Addis Ababa, Ethiopia

\* zenebe.negash@aau.edu.et

## Abstract

### Background

Diabetes mellitus (DM) patients are at increased risk of developing drug therapy problems (DTPs). The patients had a variety of comorbidities and complications, and they were given multiple medications. Medication therapy management (MTM) is a distinct service or group of services that optimize therapeutic outcomes for individual patients. The study assessed the impact of provision of MTM service on selected clinical and humanistic outcomes of diabetes patients at the diabetes mellitus clinic of Tikur Anbessa Specialized Hospital (TASH).

### Methods

A pre-post interventional study design was carried out at DM clinic from July 2018 to April 2019. The intervention package included identifying and resolving drug therapy problems, counseling patients in person at the clinic or through telephone calls, and providing educational materials for six months. This was followed by four months of post-intervention assessment of clinical outcomes, DTPs, and treatment satisfaction. The interventions were provided by pharmacist in collaboration with physician and nurse. The study included all adult patients who had been diagnosed for diabetes (both type I & II) and had been taking anti-diabetes medications for at least three months. Patients with gestational diabetes, those who decided to change their follow-up clinic, and those who refused to participate in the study were excluded. Data were analyzed using Statistical Package for the Social Sciences (SPSS). Descriptive statistics, t-test, and logistic regressions were performed for data analyses.

### Results

Of the 423 enrolled patients, 409 fulfilled the criteria and included in the final data analysis. The intervention showed a decrease in average hemoglobin A1c (HbA1c), fasting blood

**Data Availability Statement:** All relevant data are within the manuscript and its Supporting Information files.

**Funding:** This study received funding support from Addis Ababa University (AAU) Medication Therapy Management (MTM) thematic research project grant and Office of the Graduate Programs, AAU. The funders had no role in study design, data collection and analysis, decision to publish, or preparation of the manuscript.

**Competing interests:** The authors have declared that no competing interests exist.

sugar (FBS), and systolic blood pressure (SBP) by 0.92%, 25.04 mg/dl, and 6.62 mmHg, respectively ($p<0.05$). The prevalence of DTPs in the pre- and post-intervention of MTM services was found to be 72.9% and 26.2%, respectively ($p<0.001$). The overall mean score of treatment satisfaction was 90.1(SD, 11.04). Diabetes patients of age below 40 years (92.84 (SD, 9.54)), type-I DM (93.04 (SD, 9.75)) & being on one medication regimen (93.13(SD, 9.17)) had higher satisfaction score ($p<0.05$).

## Conclusion

Provision of MTM service had a potential to reduce DTPs, improve the clinical parameters, and treatment satisfaction in the post-intervention compared to the pre-intervention phase.

## Introduction

Globally, an estimated 422 million adults were living with diabetes mellitus (DM) in 2014, with prevalence of 8.5% in the adult population, which is expected to rise to 641.8 million by 2040 [1]. An estimated 14.2 million adults in Africa had diabetes, with a regional prevalence of 2.1–6.7%. In Ethiopia about 2.6 million adults had diabetes in 2017 [2]. Global health spending to treat diabetes and prevent complications accounted for 11.6% of the total health expenditure in 2015 and this value was between $80–200 millions in Ethiopia [1].

Based on etiopathogenesis of the disease, diabetes can be classified as type I diabetes(5 to 10%), type II diabetes($\sim$ 90–95% of diabetes), gestational diabetes mellitus (GDM) and monogenic diabetes and secondary diabetes(less common) [3]. The management of diabetes depend on understanding of the pathophysiology of the disease. In type I diabetes mellitus, there is a significant insulin deficiency and the only therapeutic option is the administration of insulin or insulin analog. Metformin in combination with insulin may help type I DM patients who are overweight, taking high doses of insulin, or have a HbA1c of more than 8%, according to recent research [3,4].

While in type II diabetes mellitus individuals, have relative insulin deficiency and peripheral insulin resistance that require either oral medication or insulin/insulin analog or both. The currently available class of oral antidiabetes medication include sulphonylurea(SUs) and metiglinides, which function as insulin secretagogues and promote insulin secretion directly; biguanides (such as metformin) and thiazolidinediones (TZD), which improve insulin sensitivity; and alpha glucosidase inhibitors, which minimize the need for post-prandial insulin secretion by slowing intestinal carbohydrate absorption [2].

The management of diabetes is complex, requiring more than plasma glucose control. It comprises managing DM-related complications and modifying risk factors [5]. The management involves combining lifestyle modification with the pharmacological agents to address the multiple pathophysiological defects [6]. Due to coexistence of comorbidities and complications, DM management is challenged by the occurrence of drug therapy problems (DTPs). DTPs can occur at different stages of medication use starting from prescriber to patients and often deleterious and costly [7–9]. DTPs are significant public health issue worldwide and have been significantly increasing overtime [1,10].

Studies conducted in different parts of the world showed that DTPs are highly prevalent. Those carried out in Asia and Africa reported an average of 1–3 DTPs per patient [10–13]. A similar rate of DTPs has been reported by various studies conducted in Ethiopia [14–16]. Drug-related hospital admissions are also significantly increasing overtime. It account for

5–10% of admissions and more than 50% of which were avoidable [17]. In the USA, an esti-
mated 100,000 deaths occur annually due to DTPs, costing taxpayers approximately $201.4 bil-
lion per annum [18].

To reduce such drug therapy issues and attain targeted therapeutic outcomes, implementa-
tion of medication therapy management (MTM) service is crucial. MTM is a pharmacist-pro-
vided standard practice for assessing patient's drug-related needs as well as identifying and
resolving DTPs.

MTM service involves providing self-management education, addressing medication
adherence issues, and considering preventative health strategies to optimize therapy and
improve clinical outcomes [8]. It begins by comprehensive medication review to ensure if the
patient's medication-related needs have been met and all of his/her medications are appropri-
ate, effective, safe, and convenient. At the end of the review, a care plan is developed and
shared with the patient and the primary care provider to resolve and prevent any DTPs [7,19].

A meta-analysis of 44 studies assessing the effectiveness of MTM services in patients with
chronic diseases showed that MTM improved prescribing, use and adherence related issues
[20]. The Asheville project showed that more than 50% of patients achieve optimal hemoglo-
bin A1c (HbA1c) at each follow-up assessment, indicating the long-term clinical and eco-
nomic gains [21]. In a prospective pre-post longitudinal study, HbA1c levels decreased on
average by 0.27, while systolic and diastolic blood pressure decreased by 6.0 and 4.2 mmHg,
respectively [22]. In a pragmatic randomized controlled trial of telephonic MTM to reduce
hospitalization in home health patients, the intervention group was three times less likely to be
hospitalized compared to the usual care group [23]. Even though many studies are available on
the implementation of MTM service in different countries, no study exists about this service in
diabetes patients of Ethiopia. Thus, the aim of this study was to assess the impact of introduc-
tion of MTM service at the diabetes clinic of Tikur Anbessa Specialized Hospital (TASH).

## Methods

### Design and patients

A pre-post interventional study was carried out from 6 July 2018 to 30 April 2019 (6-month
intervention and 4-month assessment) to assess the effectiveness of MTM in patients attending
the diabetes clinic of TASH. TASH is the largest referral hospital that offers a comprehensive
health care service for more than 500,000 patients per year through its 20 specialty clinics and
5 main inpatient service departments. The diabetes clinic provides outpatient service for 6,000
adult patients annually, with an average of about 250 patients per week.

All adult patients diagnosed with DM (both type I &II) and on anti-diabetes drugs for at
least three months in diabetes clinic of TASH were included in the study. Gestational DM
patients, patients planning to change the follow up clinic, and patients unwilling to participate
in the study were excluded. The study was approved by the Institutional Review Board (IRB)
of the School of Pharmacy, Addis Ababa University (Ref. No.: 002/17/SPharma). Written
informed consent was obtained from participants before collecting the required data.

### Sampling

A sample size(n) of 423 were computed based on single proportion formula assuming DTPs
prevalence(p) 50%. A critical value for normal distribution at 95% confidence interval, z-value
of 1.96 was used with a margin of error(d) 5% in sample size calculation.

$$n = \frac{Z^2 p(1-p)}{d^2} = \frac{(1.96)^2 x(0.5)(0.5)}{(0.05)^2} = 384.16$$

Therefore, with adjustment for 10% contingency (for non-response), the total sample size were 423.

During the three months recruitment periods, patients who had an appointment at the diabetes clinic of TASH were made the sampling frame and the sampling fraction was calculated. The average 125 daily attendees were used for calculation of constant (k). The total sample size (423) was divided by the number of days the clinic provide service within three months (24 days) of the recruitment period to get the estimated sample of participants per day. Based on this calculation about 18 patients were sampled each day. This was made for the purpose of participant distribution throughout the study period for better representativeness. A systematic random sampling technique was used based on list of patients' appointment record by calculating sampling interval as K = N/n (where N (125) average number of patients per day; n (18) is sample to be taken per day). Then participant's medical card number (ID) were taken every seventh interval for comprehensive chart review. A colored sticker was posted on the patient chart to identify easily during follow up and to avoid double recruitments.

## Intervention

On the day of appointment, recruited patients were interviewed for additional information (social habit, economic status, educational and marital status, physical activity etc) and provided with MTM services after they got usual care from the primary physician. Based on the standard diabetes treatment guideline [2,24] trained pharmacists reviewed patients' medication regimen and rendered verbal education and training on medication use and best administration sites with the goal of optimizing medication therapy. The interventions also involved identifying DTPs, such as medication duplications, drug interactions, dosing for renal and liver impairment, suspected adverse drug reactions (ADRs), therapeutic drug monitoring and inappropriate non-pharmacological managements. DTPs were identified and classified using the Cipolle's [7] tools.

The pharmacists also provided brochures prepared in a local language (Amharic) as intervention package to increase patients' awareness about their disease condition and lifestyle modification. The package also included delivering personal medication data book comprising personal information, personal medication record (PMR), medication action plan (MAP), investigation value recorder (blood pressure (BP), fasting blood sugar (FBS), random blood sugar (RBS) and HbA1c) as well as additional information about hypoglycemia symptoms and its management.

In DTPs identification, age, comorbidities and complications, glycemic control, drug safety profile and proper drug selection, dosage titration, indication for therapy, untreated indication, clinical characteristic, and organ function tests were considered. After reviewing, the pharmacists provided recommendations after consultation with the treating physician and patients. They then documented any interventions provided during each visit and made follow up through telephone calls. Interventions required during the follow-up were made following consultations with the treating physicians.

For both pre- and post-MTM assessment, a recent HbA1c and an average consecutive FBS were considered. Time spent with each patient was 15–20 min for interview as well as for medication review, and 15–20 min for intervention, patient medication record, and documentation. All patients recruited for the intervention were used for post-MTM assessment study.

## Outcome measures

The main outcome measured were the change in DTPs, clinical and humanistic (treatment satisfaction) outcomes from pre-MTM (baseline) to post-MTM after intervention. The clinical

outcomes include glycemic control (HbA1c, FBS, RBS), blood pressure, lipids profile (HDL, LDL, triglycerides) levels. DTP were determined by modified Cipolle [7] while the humanistic outcome treatment outcome were measured by satisfaction with medicines questionnaire (SATMED-Q) [25,26].

Hyperglycemia was defined as an average FBS level of above 130 mg/dl for patients between 18 and 60 years old with no comorbid with disease duration of below 8years and values above 150 mg/dl for those above 60 years of age and patients with multiple comorbid and also those with disease duration more than 8 years. Controlled hypertension: if patient diagnosed as hypertension and initiated with medication or lifestyle modification achieve therapeutic goal of SBP/DBP< 130-140/90 mmHg. Controlled lipid profile: if dyslipidemic patients achieve lipid profile of HDL-C >40 mg/dL, LDL-C <100 mg/dL, TG <150 mg/dL, and TC <200mg/dL [2,24,27].

The most commonly used HbA1c goal for many nonpregnant adults is less than 7% (53 mmol/mol). The more stringent HbA1c goals (such as less than 6.5% [48 mmol/mol]) were also used for selected individual patients if it can be achieved without significant hypoglycemia or other adverse effects of treatment (i.e., polypharmacy). Appropriate patients might include those with short duration of diabetes, type 2 diabetes treated with metformin only, long life expectancy, or no significant cardiovascular disease. In addition less stringent HbA1c goals (such as less than 8% [64 mmol/mol]) were also considered for patients with a history of severe hypoglycemia, limited life expectancy(eg, <10 years), advanced microvascular or macrovascular complications, extensive comorbid conditions, or long-standing diabetes in whom the goal is difficult to achieve despite diabetes self-management education, appropriate glucose monitoring, and effective doses of multiple glucose-lowering agents including insulin [2,24,27].

## Data collection and management

Data were collected using a pre-tested data abstraction format, modified Cipolle DTP identification tools[7], and satisfaction with medicines questionnaire (SATMED-Q) [25,26] as instruments. In DTP assessment, categories (1–6) associated with indication, effectiveness, and safety were used (Appendix I in S1 File). However, the seventh category that assesses medication adherence was removed, as it had to be assessed by Morisky medication adherence scale (MMAS) tool. During reporting each value (frequency) was recorded as prevalence of the specific category and then converted to percentage.

A self-administered SATMED-Q questionnaire was used to measure patients' treatment satisfaction in persons with any chronic disease treated with medicines (Appendix I in S1 File). It is a brief, feasible and easy to self-administer that has 17 items, assessing six treatment satisfaction domains; undesirable side effects (3 items), treatment effectiveness (3 items), convenience of use (3 items), impact on daily activities (3 items), medical care (2 items) and global satisfaction (3 items) each of which is computed as a score. Each item in the scale uses a five-point Likert scale (not at all (0), a little bit (1), some-what (2), quite a bit (3), very much (4)); overall and domain scores range from zero to 68, with higher scores indicating greater levels of treatment satisfaction. The resultant total composite score could be transformed to a more intuitive and easier to understand metric with a minimum of 0 and a maximum of 100, using the following expression

$$Y' = \frac{Yobs - Ymin}{Ymax - Ymin} * 100 = Yobs*1.471$$

Where Y max is 68 (maximum total score), Ymin is zero (minimum total score), Yobs is the total patient score, and Y' is the transformed score. A similar expression can be used to change the metric of each individual domain [25,26].

Demographic (age, sex, marital status, education, residence, and occupation status) and clinical information (disease type, duration of the disease, comorbidity & complication, type of medications, source of medication, and lifestyle (physical activity, alcohol use, smoking and dietary status)) were collected using the data abstraction format (Appendix I in S1 File). Online resources (Micromedex, Launch Lexi-Interact™) and Standard textbooks [28,29] were used for DTP identification. Other variables, including weight, height, body mass index (BMI), waist circumference (WC), BP were also recorded.

For the intervention phase, two clinical pharmacists and two nurses were recruited and provided with a 2-day intensive theoretical and practical training on the procedure followed during intervention, intervention implementation, and mitigation of challenges. For the post-intervention assessment, another two nurses and two pharmacists were recruited and trained for data collection based on their interest and full commitment to the MTM project. The role of the pharmacists was identifying DTPs and providing intervention package, while the nurses were involved in facilitating and coordinating activities during intervention and assessment.

### Data analysis

Data were checked, cleaned, entered and analyzed using SPSS version 25. Descriptive analysis was computed as frequency and percent for categorical variable, and mean and standard deviation (SD) for continuous variables. To examine the influence of different variables on DTPs logistic regression analysis was used. To control potential confounders, the variables in bivariate analysis with $p$-value $\leq 0.20$ were further analyzed in multivariate logistic regression. Paired sample t-test was used to compare the difference between the mean of pre- and post-intervention continuous variables while McNemar test was for categorical variables. Association between treatment satisfaction (mean scores of SATMED-Q) and socio-demographic and clinical characteristics was determined using independent t-test for mean values of two continuous variables and one-way analysis of variance (ANOVA) with post hoc analysis for mean values of more than two continuous variables. A 95% CI and p-value of <0.05 was considered statistically significant for all data analysis.

## Results

### Socio-demographic and clinical characteristics of patients

Of the 423 patients recruited, 14 were excluded as per the eligibility criteria. Two patients changed their follow up site, one patient became pregnant, and eleven patients refused to participate in the post-intervention assessment. Patients had a mean age of 52.3(SD, 15.6) years and most (42.5%) were in the age range of 40–60 years. Majority of them were females (54.5%), married (71.4%) and resident of Addis Ababa (84.4%) (Table 1).

Greater proportion (78.2%) of the patients had type-II DM. Comorbidity and complication were found in 73% (299) and 37% (151) of the study participants, respectively. Hypertension (56.2%) and neuropathy (30.1%) were the two most common comorbidities and complications, respectively (Table 1).

Majority (77%) of the patients had 1–2 pharmacist visits and the rest had three or more visits. McNemar test revealed a significant decrease (p<0.001) in the proportion of hyperglycemic patients from 68% in the pre-intervention to 40.3% in the post-intervention phase. Likewise, hypoglycemia occurrence came down from 18.3% to 5.6% (p<0.001). The intervention also produced an improvement in the clinical characteristics of patients, including BMI, WC, BP, HbA1c and FBS. Based on the paired sample t-test, significant improvement (p<0.05) was noted in the post-intervention values for SBP, HbA1c, and FBS compared to the corresponding pre-intervention values (Table 2).

**Table 1. Socio-demographic and clinical characteristics of adult patients.**

| Variables | Categories | N (%) |
|---|---|---|
| Age(Years) | Mean ± SD | 52.3 ±15.6 |
| | < = 40 | 99(24.2) |
| | 40–60 | 174(42.5) |
| | >60 | 136(33.3) |
| Sex | Male | 186(45.5) |
| | Female | 223(54.5) |
| Marital status | Married | 292(71.4) |
| | Single | 69(16.9) |
| | Divorced | 19(4.6) |
| | Widowed | 29(7.1) |
| Education | Unable to write & read | 28(6.8) |
| | Informal education | 25(6.1) |
| | Primary school | 74(18.1) |
| | Secondary school | 131(32.0) |
| | Diploma and above | 151(36.9) |
| Residence | Addis Ababa | 345(84.4) |
| | Out of Addis Ababa | 64(15.6) |
| Occupational status | Employed | 115(28.1) |
| | Unemployed | 82(20.1) |
| | self-employed | 59(14.4) |
| | Others[a] | 153(37.4) |
| Source of medication | Buying | 78(19.1) |
| | Free | 331(80.9) |
| Allergy to any medication | Known | 29(7.1) |
| | No/Not known | 380(92.9) |
| Social drug use | X-smoker | 4(1.0) |
| | Smoker | 6(1.5) |
| | Alcohol consumption | 59(14.4) |
| | Caffeine intake | 280(68.5) |
| | Khat chewing | 10(2.4) |
| Type of physical activity | No | 56(13.7) |
| | Walking | 193(47.2) |
| | Exercise | 31(7.6) |
| | Daily activity | 129(31.5) |
| Family history | No/Unknown/ Other[b] | 349(85.3) |
| | Mother/Father/Sister/Brother | 60(14.7) |
| Type DM | Type I | 89(21.8) |
| | Type II | 320(78.2) |
| Number of comorbidities | No | 110(26.9) |
| | 1–2 | 240(58.7) |
| | > = 3 | 59(14.4) |
| Types of comorbidity | Hypertension | 230(56.2) |
| | Dyslipidemia | 96(23.5) |
| | IHD | 52(12.7) |
| | CKD | 26(6.4) |
| | RVI | 15(3.7) |
| | Asthma | 9(2.2) |
| | Others[c] | 105(25.7) |

(*Continued*)

**Table 1.** (Continued)

| Variables | Categories | N (%) |
|---|---|---|
| Number of complications | No | 258(63.1) |
| | 1–2 | 141(34.5) |
| | > = 3 | 10(2.4) |
| Types of complications | Neuropathy | 123(30.1) |
| | Nephropathy | 30(7.3) |
| | Retinopathy | 29(7.1) |
| | Diabetic foot ulcer | 5(1.2) |
| | Others[d] | 14(3.4) |
| Number of Medications | Mean ± SD | 4.3±2.4 |
| | One | 76(18.6) |
| | Two-Four | 137(33.5) |
| | Five and above | 195(47.7) |
| Duration of diabetes (Years) | Mean ± SD | 13.9± 8.6 |
| | <5 | 41(10.0) |
| | 5–10 | 59(14.4) |
| | 10–15 | 69(16.9) |
| | > = 15 | 129(31.5) |

[a]Retired

[b]Grandparents, Relative

[c]Thyroid disorders, osteoarthritis, psychotic disorder, infection, cancer, seizure, obesity

[d]Peripheral arterial disease, autonomic gastroparesis; CKD: Chronic kidney disease; IHD: Ischemic heart disease; RVI: Retroviral infection; SD: Standard Deviation.

Patients received an average of 1.49 anti-diabetes medications per patient. Insulin (41.3%) was the most prescribed medication followed by Metformin + Insulin (29.6%). Apart from anti-diabetes medications, antihypertensive agents (53.8%) were also the predominantly prescribed medications. Among the total patients, 18.6% used one, 33.5% used 2–4, and 47.7% used ≥5 medications.

## Drug therapy problems

During the pre-intervention phase, 578 DTPs were identified in 298 (72.9%) patients, with a mean of 1.94 (SD, 1.06) DTPs per patient. Of this, one DTP occurred in 130 (43.6%) and two DTPs in 92 (30.9%) patients (Fig 1). In the post-intervention phase, the number of DTPs was reduced to 128 and identified in 107 (26.2%) patients (p<0.001) (Table 3). The most frequent type of DTP was ADRs. ADRs occurred in about 38.2% (221) of patients and were undesired in 25.3% (n = 146) of the cases. The second most commonly encountered DTP was needs additional drug therapy (26.5%, 152) followed by dosage too low (25.4%, 147). Among anti-diabetes drugs used for management, insulin (41.3%) was the most frequently involved in DTPs followed by the combination of metformin and insulin (32.1%). Statin (65.2%) and angiotensin converting enzyme inhibitors (ACEI)/ angiotensin-receptor blockers(ARBs)(49.8%) were the most common drugs involved in DTPs among other class of drugs.

Once the DTPs were identified, discussions were held with physicians (about the treatment regimen), nurses (about patient education), pharmacists (about dispensing the medications), and patients (about adherence to interventions). The clinical pharmacists made interventions as appropriate and the acceptance rate was 86.3%.

**Table 2. Clinical measures for mean scores and paired samples t-test for clinical outcome measures among adult diabetic patients.**

| Variables | Descriptive Statistics (Mean ± SD) | | Paired Differences Mean±SD | 95% CI of the Difference | | t | p-value |
|---|---|---|---|---|---|---|---|
| | Pre MTM | Post MTM | | Lower | Upper | | |
| BMI | 25.0±4.3 | 24.7±4.3 | 0.13±1.6 | -0.03 | 0.28 | 1.61 | 0.108 |
| WC | 36.8±5.7 | 35.7±6.7 | -1.53±10.55 | -2.56 | -0.51 | -2.94 | 0.003 |
| SBP | 141.2±18.7 | 134.5±15.9 | 6.62 ± 26.75 | 4.02 | 9.22 | 5.01 | 0.000 |
| DBP | 80.9±10.5 | 79.3±9.3 | -2.85±16.76 | 0.83 | -4.48 | -3.441 | 0.001 |
| HbA1c | 9.3±1.7 | 8.2±1.6 | 0.92±3.04 | 0.63 | 1.22 | 6.13 | 0.000 |
| FBS | 167.0±61.4 | 141.7±47.5 | 25.04±62.93 | 18.93 | 31.16 | 8.05 | 0.000 |

Key: BMI: Body mass index; DBP: Diastolic blood pressure; FBS: Fasting blood sugar; HbA1c: Hemoglobin A1c; MTM: Medication therapy management; SBP: Systolic blood pressure; SD: Standard deviation of mean; WC: Waist circumference.

Logistic regression analysis was performed to identify predictors of DTPs (Table 4). From the socio-demographic and clinical characteristic incorporated in the analysis, source of medication and educational status were significantly associated with DTPs

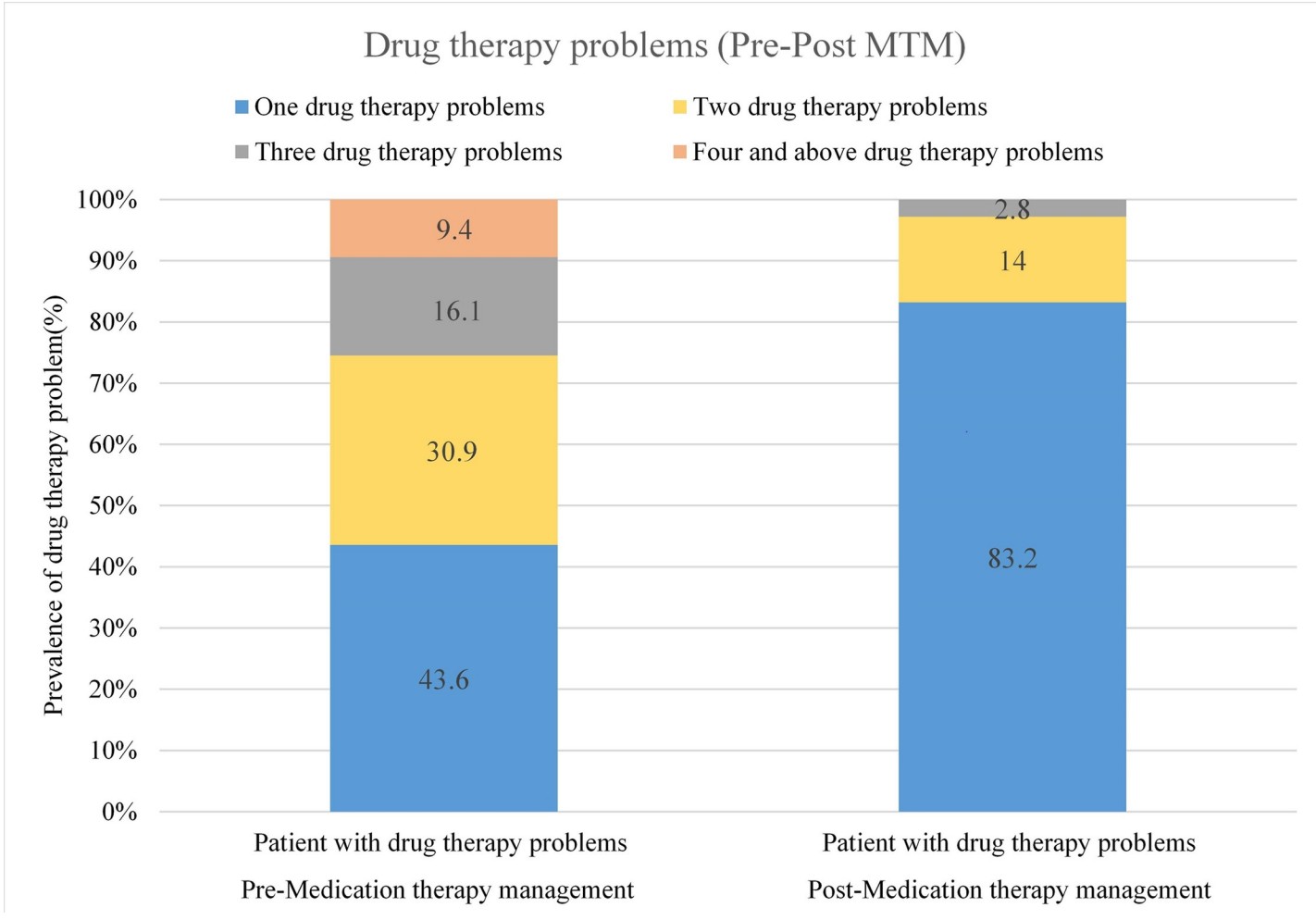

**Fig 1. Drug therapy problems among adult patients with diabetes.**

**Table 3. Drug therapy problems and causes among adult patients with diabetes.**

| Types of drug therapy problems | Causes | Participants | | p-value* |
|---|---|---|---|---|
| | | Pre-MTM (N(%)) | Post-MTM (N(%)) | |
| Unnecessary drug therapy | | 28(4.9) | 10(7.8) | <0.005 |
| | Inappropriate Duplication of drug therapy | 5(0.9) | 2(1.6) | 0.500 |
| | No medical indication at this time | 3(0.5) | 1(0.8) | 1.000 |
| | Non drug therapy more appropriate | 5(0.9) | 2(1.6) | 0.500 |
| | Addiction/recreational drug use | 12(2.1) | 4(3.1) | 0.125 |
| | Treating avoidable adverse reaction | 3(0.5) | 1(0.8) | 1.000 |
| Needs additional drug therapy | | 152(26.5) | 26(20.3) | <0.001 |
| | Preventive therapy | 59(10.2) | 12(9.4) | <0.001 |
| | Untreated condition | 43(7.4) | 8(6.3) | <0.005 |
| | Synergistic therapy | 51(8.8) | 6(4.7) | <0.001 |
| Ineffective drug product | | 6(1.0) | 0 | 0.125 |
| | More effective drug available | 4(0.7) | 0 | 0.250 |
| | Dosage form inappropriate | 2(0.4) | 0 | 1.000 |
| Dosage too low | | 147(25.4) | 19(14.8) | <0.001 |
| | Ineffective dose | 54(9.3) | 9(7.0) | <0.001 |
| | Frequency inappropriate | 32(5.5) | 4(3.1) | <0.005 |
| | Incorrect administration | 41(7.1) | 3(2.3) | <0.001 |
| | Drug interaction | 8(1.4) | 0 | 1.000 |
| | Incorrect storage | 8(1.4) | 3(2.3) | 1.000 |
| | Duration inappropriate | 4(0.7) | 0 | 1.000 |
| Adverse drug reaction | | 221(38.2) | 65(50.8) | <0.001 |
| | Undesirable side effect | 146(25.3) | 55(43.0) | <0.001 |
| | Unsafe drug for the patient | 3(0.5) | 0 | 0.500 |
| | Drug interaction | 58(10.0) | 8(6.3) | <0.001 |
| | Incorrect administration | 6(1.0) | 2(1.6) | 0.453 |
| | Allergic reaction | 5(0.9) | 0 | 0.125 |
| | Dosage increase/decrease too fast | 3(0.5) | 0 | 0.500 |
| Dosage too high | | 23(4.0) | 8(6.3) | 0.180 |
| | Dose too high | 12(2.1) | 4(3.1) | 0.219 |
| | Needs additional monitoring | 3(0.5) | 4(3.1) | 1.000 |
| | Frequency too short | 4(0.7) | 0 | 0.500 |
| | Duration too long | 4(0.7) | 0 | 0.500 |
| Number of patient with DTPs | | 298(72.9) | 107(26.2) | <0.001 |
| Number of DTPs identified | | 578 | 128 | |
| Average number of DTPs per participant | | 1.94±1.06 | 1.2±0.47 | |

N: Number of drug therapy problem; DTPs: Drug therapy problems; MTM: Medication therapy management; *McNemar test.

(p<0.05) during the pre-MTM intervention. Patients who paid for their medications had about two-fold risk of developing DTPs as compared to patients who got their medications for free (AOR = 2.27, 95% CI: 1.08–4.77). In addition, patients with primary level of education were also about three times more at risk to develop DTPs as compared to patients who had diploma and above (AOR = 2.94, 95% CI: 1.25–6.91). Following the MTM intervention the male gender was (AOR = 3.06, 95% CI: 1.54–6.07) three times more likely to develop DTPs than female gender.

**Table 4. Bivariate and multivariate analysis of factors associated with drug therapy problems among adult patients with diabetes.**

| Variables | Categories | During MTM | | Post MTM | |
|---|---|---|---|---|---|
| | | Odds Ratios (95% CI) | | Odds Ratios (95% CI) | |
| | | COR | AOR | COR | AOR |
| Age(years) | < = 40 | 1.00 | | 1.00 | |
| | 40–60 | 0.92(0.21–4.01) | 0.67(0.16–2.93) | 0.74(0.19–2.97) | 0.71(0.18–2.80) |
| | >60 | 1.37(0.26–7.19) | 1.06(0.20–5.55) | 0.75(0.15–3.77) | 0.75(0.15–3.70) |
| Gender | Female | 1.00 | | 1.00 | |
| | Male | 2.04(1.01–4.1) | 1.38(0.69–2.76) | 4.17(1.96–8.87) | **3.85(1.84–8.07)** |
| Marital status | Widowed | 1.00 | | 1.00 | |
| | Married | 2.23(0.47–10.55) | 2.15(0.48–9.62) | 0.35(0.08–1.63) | 0.34(0.08–1.56) |
| | Single | 2.28(0.28–14.97) | 2.49(0.40–15.74) | 0.31(0.05–1.92) | 0.31(0.05–1.89) |
| | Divorced | 3.02(0.32–28.66) | 3.15(0.38–25.83) | 0.70(0.06–7.5) | 0.75(0.07–7.88) |
| Educational status | Diploma and above | 1.00 | | 1.00 | |
| | Unable to write & read | 1.22(0.31–4.9) | 1.05(0.29–3.86) | 3.76(0.8–17.56) | 3.51(0.76–16.27) |
| | Informal education | 0.99(0.21–4.47) | 0.85(0.20–3.53) | 2.91(0.52–16.19) | 2.51(0.47–13.4) |
| | Primary school | 3.95(1.52–10.24) | **2.94(1.25–6.91)** | 1.5(0.59–3.82) | 2.06(0.91–2.76) |
| | Secondary school | 1.96(0.85–4.49) | 1.41(0.67–2.99) | 2.22(0.96–5.12) | 0.35(0.08–4.65) |
| Residency | Addis Ababa | 1.00 | | 1.00 | |
| | Out of Addis Ababa | 1.67(0.60–4.71) | 1.68(0.62–4.52) | 0.6(0.21–1.67) | 0.63(0.23–1.73) |
| Occupation | Employed | 1.00 | | 1.00 | |
| | Unemployed | 0.32(0.09–1.19) | 0.44(0.12–1.58) | 1.37(0.38–4.98) | 1.42(0.40–5.05) |
| | Self–employed | 0.54(0.16–1.84) | 0.80(0.26–2.50) | 0.42(0.15–1.19) | 0.46(0.17–1.27) |
| | Others[a] | 0.61(0.22–1.71) | 0.83(0.309–2.22) | 0.99(0.38–2.56) | 1.05(0.41–2.69) |
| Type DM | Type-II | 1.00 | | 1.00 | |
| | Type-I | 1.58(0.37–6.78) | 0.72(0.17–3.10) | 1.3(0.38–4.44) | 1.28(0.38–4.32) |
| Number of Comorbidities | No | 1.00 | | 1.00 | |
| | 1–2 | 0.91(0.30–2.71) | 0.84(0.30–2.38) | 1.7(0.64–4.52) | 1.65(0.63–4.32) |
| | > = 3 | 0.63(0.14–284) | 0.75(0.19–3.00) | 1.49(0.43–5.23) | 1.51(0.43–5.26) |
| Number of Complications | No | 1.00 | | 1.00 | |
| | 1–2 | 0.92(0.40–2.12) | 0.88(0.41–1.90) | 0.75(0.36–1.56) | 0.76(0.37–1.55) |
| | > = 3 | 0.25(0.16–4.01) | 0.24(0.02–3.20) | 2.23(0.19–26.15) | 2.26(0.19–26.83) |
| Duration of DM | <5years | 1.00 | | 1.00 | |
| | 5-10years | 0.70(0.19–2.62) | 0.74(0.22–2.46) | 0.49(0.15–1.58) | 0.49(0.15–1.56) |
| | 10-15years | 0.93(0.25–3.47) | 0.80(0.24–2.65) | 0.61(0.19–1.99) | 0.61(0.19–1.94) |
| | > = 15years | 0.82(0.23–2.95) | 0.82(0.26–2.62) | 0.65(0.21–2.00) | 0.61(0.20–1.84) |
| Source of medication | Free | 1.00 | | 1.00 | |
| | Buying | 2.49(1.09–5.70) | **2.27(1.08–4.77)** | 1.51(0.62–3.66) | 1.58(0.65–3.82) |

[a]Retired AOR: Adjusted odds ratio; C.I. confidence interval; COR: Crude odds ratio; MTM: Medication therapy management.

## Treatment satisfaction

Based on SATMED-Q score tool the treatment satisfaction rate was described in Fig 2 according to the domain score. The satisfaction rate ranged from 88.3 (medical care domain) to 91.6 (convenience domain). The overall mean score of treatment satisfaction was 90.1(SD, 11.04). Socio-demographic and clinical characteristics of the patients also had an effect on treatment satisfaction of patients (Table 5). Diabetic patients of younger age group (below 40 years) (92.84(SD, 9.54)), type-I DM (93.04(SD, 9.75)) and managed by one medication (93.13(SD, 9.17)) had a significantly higher treatment satisfaction score compared to their counterparts (p<0.05).

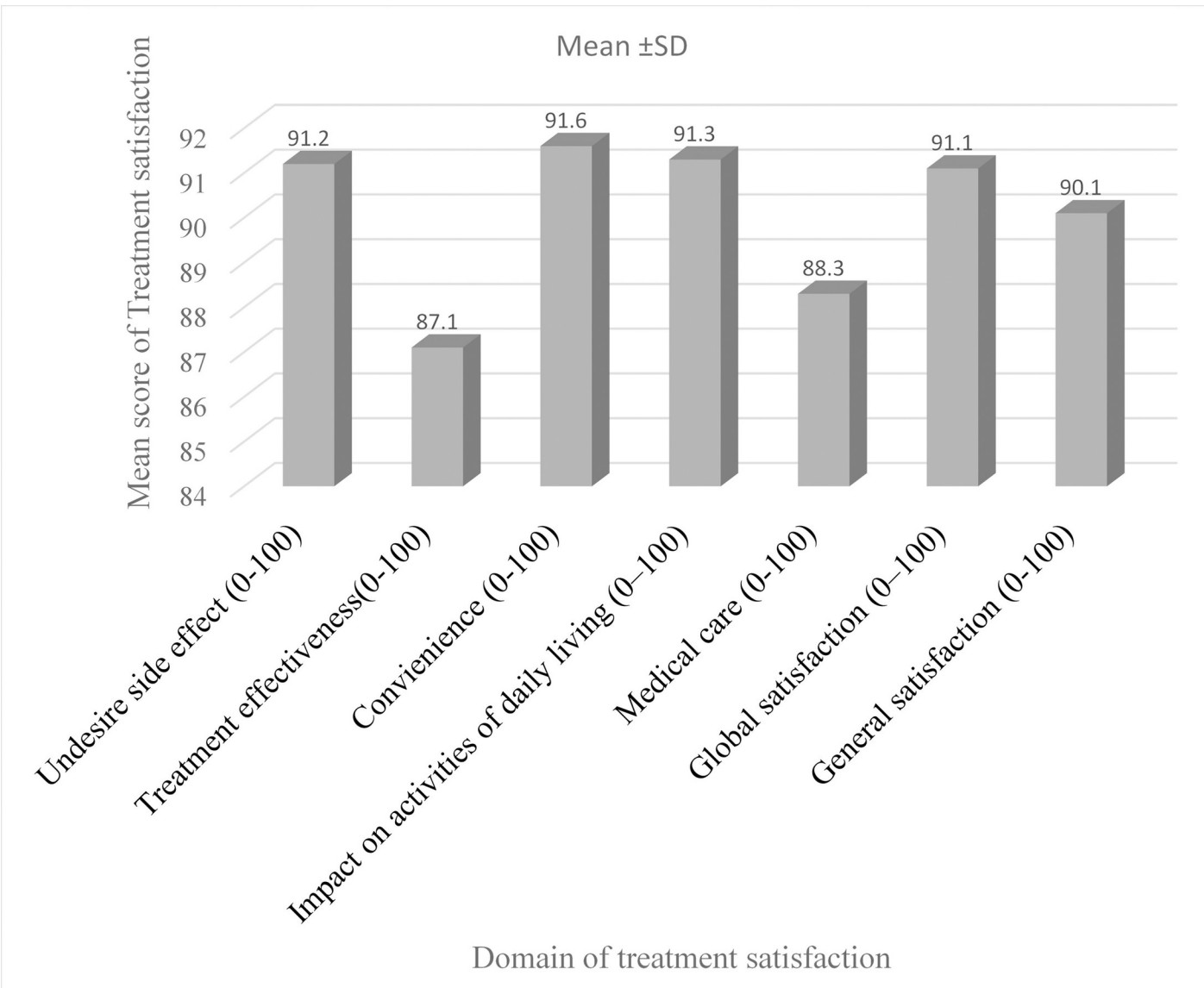

**Fig 2. Treatment satisfaction following MTM intervention among adult diabetic patients.**

## Discussion

Patients with diabetes are at high risk of having DTPs and non-adherence issues due to comorbidities and polypharmacy [30]. Identification and resolution of DTPs contributes to better clinical outcome and reduction of drug-related hospitalizations, morbidity and mortality [10]. Treatment satisfaction is also an important component of the quality of medical care [31]. Therefore, this study evaluated the impact of introducing MTM services on such issues surrounding DM patients.

### Clinical outcome

In this study co-morbidities and complications were common among study participants, probably attributed to higher mean age (52 years) and duration of illness (14 years) as well as type

**Table 5. Relationship between treatment satisfaction and different characteristics of patients with diabetes.**

| Variables | Category | N | Mean SATMED-Q score ± SD | F | P-value |
|---|---|---|---|---|---|
| Age(years) | < = 40 | 99 | 92.84± 9.54 | 3.060 | **0.048**[*] |
| | 40–60 | 174 | 88.99± 11.47 | | |
| | >60 | 136 | 89.51± 11.24 | | |
| Gender | Male | 186 | 91.87± 10.32 | 2.800 | 0.095[**] |
| | Female | 223 | 88.61± 11.43 | | |
| Marital status | Married | 292 | 89.86± 11.39 | 0.302 | 0.824[*] |
| | Single | 69 | 91.65± 9.76 | | |
| | Divorced | 19 | 88.46± 12.66 | | |
| | Widowed | 29 | 89.86± 9.22 | | |
| Type of DM | Type I | 89 | 93.04± 9.75 | 5.691 | **0.018**[**] |
| | Type II | 320 | 89.28± 11.25 | | |
| Number of Comorbidities | No | 110 | 92.16± 10.26 | 2.608 | 0.075[*] |
| | One-Two | 240 | 89.14± 11.27 | | |
| | Three and above | 59 | 90.14± 11.15 | | |
| Number of Complications | NO | 258 | 91.17± 10.53 | 2.028 | 0.133[*] |
| | One-Two | 141 | 88.64± 11.39 | | |
| | Three and above | 10 | 82.89± 14.85 | | |
| Number of medication | One | 76 | 93.13± 9.17 | 3.064 | **0.048**[*] |
| | Two-Four | 138 | 90.88± 10.83 | | |
| | Five and above | 194 | 88.47± 11.50 | | |

SATMED-Q: Satisfaction with Medicines Questionnaire; SD: Standard Deviation.

[*]One-way analysis of variance (ANOVA)

[**]Independent t-test.

of diabetes they had(most had Type 2 DM). This is consistent with studies conducted in Australia [32] and USA [33] that showed age and duration of diabetes are strongly associated with macro-vascular and microvascular events. Intervention brought about a reduction in HbA1c levels by about 0.92% from baseline and this is concordant with other numerous studies performed in pharmacist-managed ambulatory and community pharmacy diabetes care models [19,34,35]. The reduction is strongly attributed to the intervention, which targeted the optimization of medication therapy need, lifestyle modification and enhanced medication adherence through consultations. The awareness created through brochure, face-face and phone based education about the disease condition and its management; ADR prevention and management could also enhance treatment outcome [36–38].

Intervention was able to reduce proportion of patients with hyperglycemia from 68% to 40%. Similar studies done elsewhere [39,40] reported a decrement following MTM service, with varied extent depending on guideline used for cut-off points, patients' awareness to apply recommendations and other sociodemographic factors. Other clinical parameters including FBS and SBP were also reduced through the intervention as reported elsewhere [34,37,39].

## Drug therapy problems

In this study patients having at least one DTP and mean number of DTP per patient decreased after provision of intervention. This finding is concordant with earlier studies that demonstrated pharmacist-provided MTM services would result in lower DTPs prevalence [19,41,42]. This reduction was brought about by increased awareness of patients about lifestyle

modification, side effect prevention and management, medication misuse, and importance of medication adherence [43,44]. In addition, it might be also due to a good communication created between pharmacist and physician during intervention in preventing the occurrence of DTPs [37,45].

ADRs were the most common DTPs encountered as reported elsewhere [46]. The presence of multiple comorbidities and complications as well as the ensuing polypharmacy to curb them might provide an explanation for the observed abundance of ADRs. The increased proportion of elderly population as well as disease duration would also likely increase the risk of ADRs. By contrast, other studies conducted in Gondar [14] and Jimma [15] (Ethiopia), India [10], and USA [18] reported needing additional therapy and taking unnecessary drug therapy were the most prevalent DTPs. Difference in study settings, socio-demographic characteristics, recruitment criteria, pharmacist clinical skills and tools used for DTP assessment might explain the discrepancy.

Patients who paid out of pocket for their medication had a $\geq$ 2-fold risk to develop DTPs compared to those who got their medications for free. This finding is consistent with other studies showing that out of pocket cost of medication affect the medication taking behavior of patients that lead to increased DTPs [47,48]. Optimal management of diabetes, its complications and comorbidities require appropriate medication with affordable cost. If patients are unable to pay for their medications, they try to change the instruction provided or intentionally miss their medication.

Focusing on the problems identified in the pre-intervention phase, the patients were provided with constructive information and education materials. The intervention also targeted improving patients' awareness about pharmacological and non-pharmacological management of their disease condition, the importance of medication adherence and management of adverse drug events. Thus, the only factor that was still significantly associated with DTPs in the post-intervention phase was gender. The male gender was more likely to develop DTPs than the female gender. This was in line with previous studies conducted in Malaysia [12] and Jordan [49]. The likely explanation could be related to differences in health behavior as well as acceptance of recommendation between males and females [50,51].

About 86% of the interventions were accepted and implemented. ADRs, needs additional drug therapy, and dosage too low were the most addressed DTPs. Mining the literature showed a varied range of acceptance rate from 50–55% [10,52] through 70–90% [14,53,54] to 100% [35]. This relatively high acceptance rate reported in the present study might be related to the presence of many comorbidities, complications, and associated polypharmacy that called for teamwork to better manage these conditions.

## Treatment satisfaction

The overall mean score of treatment satisfaction was 90.1(SD, 11.04). This was relatively higher than the baseline satisfaction score (80.81(SD, 8.58)) [16]. This finding is consistent with other studies that showed MTM provided by pharmacist had an average score of treatment satisfaction more than 75 composite score [35,55].

Patients with Diabetes below 40 years had high treatment satisfaction score, which is congruent with studies conducted in Qatar [56] and Pakistan [57]. However, it is in stark contrast with the Netherlands study that reported young age is associated with low treatment satisfaction [31]. The lower treatment satisfaction among the elderly might be due to the presence of many comorbidities, complications and long duration of disease. This condition might lead to polypharmacy and economic burden that affects quality of life.

Type-I DM patients had high treatment satisfaction score as compared to type II DM. In contrast to our finding, in the Qatar study [56], type-I DM patients had low treatment satisfaction score. This difference might be due to difference in tool used for assessment, awareness of participants, and perception of participant towards their medication.

There are some limitation in this study. The intervention period is relatively short (six months) which may overestimate the outcome. The study is a single center study that might not allow to make generalization to the whole population. The study also lacks economic evaluation of MTM service due to resource and time limitations. To confirm the current finding and reduce the bias conducting an RCT is important. However, despite the limitations, the study could still provide some tangible evidence about effect of MTM delivery on clinical outcomes, drug therapy problem, and treatment satisfaction in developing nation with limited resource and poor facility.

## Conclusion

The study suggest that the provision of MTM service to patients with diabetes, with/without complications and comorbidity had the potential to improve the clinical parameters such as HgA1c, FBS and BP. The study also demonstrate that pharmacist led MTM service reduced the DTPs identified among ambulatory patients with diabetes in TASH. Recruiting patients to MTM service also increased treatment satisfaction among study participants.

## Supporting information

**S1 File.**
(DOCX)

## Acknowledgments

The authors would extends their special thanks to all the data collectors, participants, staff members of the diabetes clinic of Tikur Anbessa Specialized Hospital and biostatisticians for their contributions to do this research.

## Author Contributions

**Conceptualization:** Zenebe Negash, Alemseged Beyene Berha, Workineh Shibeshi, Abdurezak Ahmed, Minyahil Alebachew Woldu, Ephrem Engidawork.

**Data curation:** Zenebe Negash, Alemseged Beyene Berha, Workineh Shibeshi, Abdurezak Ahmed, Minyahil Alebachew Woldu, Ephrem Engidawork.

**Formal analysis:** Zenebe Negash, Alemseged Beyene Berha, Workineh Shibeshi.

**Funding acquisition:** Zenebe Negash, Alemseged Beyene Berha, Minyahil Alebachew Woldu, Ephrem Engidawork.

**Investigation:** Zenebe Negash, Alemseged Beyene Berha.

**Methodology:** Zenebe Negash, Alemseged Beyene Berha, Workineh Shibeshi, Abdurezak Ahmed, Minyahil Alebachew Woldu, Ephrem Engidawork.

**Project administration:** Ephrem Engidawork.

**Resources:** Alemseged Beyene Berha, Ephrem Engidawork.

**Software:** Zenebe Negash, Alemseged Beyene Berha.

**Supervision:** Alemseged Beyene Berha, Workineh Shibeshi, Abdurezak Ahmed, Minyahil Alebachew Woldu, Ephrem Engidawork.

**Writing – original draft:** Zenebe Negash, Alemseged Beyene Berha, Workineh Shibeshi.

**Writing – review & editing:** Abdurezak Ahmed, Minyahil Alebachew Woldu, Ephrem Engidawork.

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
