## [Decision Letter · Decision Letter 0]

11 Mar 2021

PONE-D-20-27442

Patient outcomes and satisfaction improves following introduction of Medication therapy management service at the ambulatory diabetes clinic of Tikur Anbessa Specialized Hospital, Addis Ababa, Ethiopia

PLOS ONE

Dear Wakjira Zenebe,

Thank you for submitting your manuscript to PLOS ONE. After careful consideration, we feel that it has merit but does not fully meet PLOS ONE’s publication criteria as it currently stands. Therefore, we invite you to submit a revised version of the manuscript that addresses the points raised during the review process. Kindly ensure the manuscript is thoroughly copyedited for any language errors in addition to the comments below.

We look forward to receiving your revised manuscript.

Kind regards,

Professor Kwasi Torpey, MD PhD MPH

Academic Editor

PLOS ONE

Journal Requirements:

2. In your Methods section, please provide additional information about the participant recruitment method and the demographic details of your participants. Please ensure you have provided sufficient details to replicate the analyses such as: a) a statement as to whether your sample can be considered representative of a larger population, b) a description of how participants were recruited, and c) descriptions of where participants were recruited and where the research took place.

3. Please include a copy of the original language version of the survey or questionnaire used in the study to ensure that you have provided sufficient details that others could replicate the analyses. Please include a copy, in both the original language and English, as Supporting Information.

Reviewers' comments:

Reviewer's Responses to Questions

**Comments to the Author**

1. Is the manuscript technically sound, and do the data support the conclusions?

Reviewer #1: Yes

Reviewer #2: Yes

2. Has the statistical analysis been performed appropriately and rigorously? 

Reviewer #1: I Don't Know

Reviewer #2: Yes

3. Have the authors made all data underlying the findings in their manuscript fully available?

Reviewer #1: Yes

Reviewer #2: Yes

4. Is the manuscript presented in an intelligible fashion and written in standard English?

Reviewer #1: Yes

Reviewer #2: Yes

5. Review Comments to the Author

Reviewer #1: Thank you very much for the chance to read your interesting and important paper. I think it will make an important contribution to the literature on medication management in diabetes patients across Africa, and beyond.

I have some comments that I would like to see addressed before publication, which I believe will improve the strength of the manuscript:

Abstract

One the abbreviation MTM is introduced, it should be used throughout the abstract.

The methods should describe who delivered the intervention (pharmacists)

Methods should also describe inclusion criteria – was the MTM clinic for all people with DM, or just Type 2? What about gestational diabetes? In short – define the type(s) of diabetes patients included

Introduction

The introduction would benefit from making distinctions between types of diabetes – epidemiology and management differs by type and this should be reflected in the opening 2 paragraphs.

Methods

Again, the authors need to describe whether the study population included type 1 and type 2 patients. It is here we learn that those with gestational diabetes were excluded.

Page. 9 HgA1c should be HbA1c – also on page 12, 19, 23

Results

In table 1 we finally learn that both type 1 and type 2 patients are included in this study. This is too late – it should be clear from the abstract and the introduction needs to address the differences in type 1 and type 2 epidemiology and management.

In table 2, the column containing ‘paired differences – mean’ is inconsistent. The change in mean appears to negative for every variable, yet only two (WC, DBP) of them are presented as negative values. I suggest this table is checked carefully for consistency

Discussion

I would encourage the authors to be more circumspect in their language about causation e.g. “The reduction is obviously ascribed to the intervention, which provided targeted optimization of medication therapy need, lifestyle modification and enhanced medication adherence through consultations.” – as this is not a randomized controlled trial (RCT), I would use more tentative language – e.g. ‘our findings strongly suggest that the intervention reduces HbA1c’. This applies to all of the reductions/positive changes that the study reports on.

The authors should note the need for an RCT to confirm findings in the limitations section.

Reviewer #2: Title: Patient outcomes and satisfaction improves following introduction of Medication therapy management service at the ambulatory diabetes clinic of Tikur Anbessa Specialized Hospital, Addis Ababa, Ethiopia

Manuscript Number: PONE-D-20-27442

Article Type: Research Article

The authors have done well. However, the following corrections should be made to make the work publishable and of good standard:

1. The title, Patient outcomes and satisfaction improves following introduction of Medication therapy management service at the ambulatory diabetes clinic of Tikur Anbessa Specialized Hospital, Addis Ababa, Ethiopia”, is a biased one from the onset. The word improves shows bias. The title should rather be, “Impact of Medication therapy management service on Patient outcomes and satisfaction in the ambulatory diabetes clinic of Tikur Anbessa Specialized Hospital, Addis Ababa, Ethiopia”

2. Again, satisfaction is an outcome (humanistic outcome), so it is like a repetition. The authors could modify the title further to ‘sellected clinical and humanistic outcomes’

3. The final title should now be,”Impact of Medication Therapy Management service on sellected Clinical and Humanistic outcomes in the ambulatory diabetes Patients of a Specialist Hospital in Addis Ababa, Ethiopia” or ,”Impact of Medication Therapy Management service on sellected outcomes in the ambulatory diabetes Patients of a Specialist Hospital in Ethiopia”

4. The fiest sentence at the background section of the abstract is rather long and passive. It should be split into two and the phrase, ‘as they often’ should be exponged. Line 17

5. The third sentence of the background section; (Thus aim of present study was to assess .....) could be rephrased to, (The study assessed .......) Line 19

6. In lines 19- 21, the objective of the study was incomplete eg he impact of MTM on what? Patient outcomes, satisfaction, etc

7. The outcome measures should be clearly delineated because patient outcomes comprises : clinical, economic, and humanistic outcomes. Here, i guess the authors are refaring to clinical outcomes , they should be specific on the them for precision

8. Line 66 should mark the begining of a new paragraph; (It also involves .....) should translate to (Also, it involves .....)

9. The phrase in line 103 (so as double recruitment ....) should be rephrased

10. In line 133, indicate the major modifications made to the questionnaire and reference it

11. How was bias handled/eliminated from the study

12. The authors should clearly state the outcome measures of this study under a sub-title at the methods section of the manuscript

13. In line 106, At the day of appointment, should be (On the day of appointment,)

14. The tables titles were rather to long. The authors should rephrase/restructure the tables titles to be very precise/concise; the name and location of the hospital can be deleted since it is already known from the title of the study.

15. Line 246-251 need additional references to establish the statements made.

16. In line 153, the statement (About 73% had) is undefined. The authored should define this population and be definite. Eg 75% of patients, type 2 diabetic patients, ???

17. Lines 271-272 should be elaborated with relevant studies/references

18. In line 319 (The intervention period is relatively short) should be corrected

19. Line 298- 303, 319- 324 were poorly discussed and without any reference. The authors should improve on them

20. The study was poorly concluded in lines 325-329.

21. Page number is missing in reference NO 26, line 439. The first year (2011) on that reference should be cross checked.

22. The journal name in reference NO 28, line 446 should be cross checked.

23. Page number is missing in reference NO 36, line 473, and NO 37 line 475 respectively.

Thank you.

6. PLOS authors have the option to publish the peer review history of their article (what does this mean?). If published, this will include your full peer review and any attached files.

Reviewer #1: No

Reviewer #2: **Yes: **Ogbonna Brian Onyebuchi

---

## [Editor Report · Decision Letter 1]

26 Apr 2021

PONE-D-20-27442R1

Impact of Medication Therapy Management service on selected Clinical and Humanistic outcomes in the ambulatory diabetes Patients of Tikur Anbessa Specialist Hospital, Addis Ababa, Ethiopia

PLOS ONE

Dear Zenebe Wakjira,

Thank you for submitting your manuscript to PLOS ONE. After careful consideration, we feel that it has merit but does not fully meet PLOS ONE’s publication criteria as it currently stands. Therefore, we invite you to submit a revised version of the manuscript that addresses the points raised during the review process. Please address the highlighted language errors.

We look forward to receiving your revised manuscript.

Kind regards,

Professor Kwasi Torpey, MD PhD MPH

Academic Editor

PLOS ONE

Journal Requirements:

Additional Editor Comments (if provided):

The manuscript has significantly improved after addressing the editorial and reviewer comments. There are still language errors that need to be addressed (Tracked changes version)

1. Title: Please capitalize appropriately

2. Line 59 should read etiopathogenesis not ethiopathogenesis

3. Place a comma after etiopathogenesis of the disease

4. Consistent use of American spelling analog versus analogue Line 64 vs 68

5. Line 93: Delete also from the beginning of the sentence

6. Line 907: It had to be assessed not It has to be assessed. Please correct

---

## [Author Response · Author response to Decision Letter 1]

27 Apr 2021

The authors would like to express their gratitude to all of the editors and reviewers for their constructive comments.

---

## [Editor Report · Decision Letter 2]

3 May 2021

Impact of Medication Therapy Management Service on Selected Clinical and Humanistic Outcomes in the Ambulatory Diabetes Patients of Tikur Anbessa Specialist Hospital, Addis Ababa, Ethiopia

PONE-D-20-27442R2

Dear Dr. Zenebe Wakjira,

We’re pleased to inform you that your manuscript has been judged scientifically suitable for publication and will be formally accepted for publication once it meets all outstanding technical requirements.

Kind regards,

Professor Kwasi Torpey, MD PhD MPH

Academic Editor

PLOS ONE
---

## [Editor Report · Acceptance letter]

19 May 2021

PONE-D-20-27442R2 

Impact of Medication Therapy Management Service on Selected Clinical and Humanistic Outcomes in the Ambulatory Diabetes Patients of Tikur Anbessa Specialist Hospital, Addis Ababa, Ethiopia 

Dear Dr. Wakjira:

I'm pleased to inform you that your manuscript has been deemed suitable for publication in PLOS ONE. Congratulations! Your manuscript is now with our production department. 

Kind regards, 

on behalf of

Professor Kwasi Torpey 

Academic Editor

PLOS ONE